:PLOS ONE

# Primary care interventions to address physical frailty among community-dwelling adults aged 60 years or older: A meta-analysis

Stephen H. -F. Macdonald[1]*, John Travers[2,3], Éidín Ní Shé[1], Jade Bailey[4], Roman Romero-Ortuno[5,6], Michael Keyes[7], Diarmuid O'Shea[7], Marie Therese Cooney[7]

1 School of Nursing, Midwifery and Health Systems, University College Dublin, Dublin, Ireland, 2 School of Medicine, University College Dublin, Dublin, Ireland, 3 HSE Specialist Training Programme in General Practice, Trinity College Dublin, Dublin, Ireland, 4 School of Medicine, University College Dublin, Dublin, Ireland, 5 Discipline of Medical Gerontology, Trinity College Dublin, Dublin, Ireland, 6 Mercer's Institute for Successful Ageing, St James's Hospital, Dublin, Ireland, 7 St Vincent's University Hospital, Dublin, Ireland

* macdonaldez@gmail.com

**Data Availability Statement:** All relevant data are within the manuscript and its Supporting Information files.

## Abstract

### Introduction

The best interventions to address frailty among older adults have not yet been fully defined, and the diversity of interventions and outcome measures makes this process challenging. Consequently, there is a lack of guidance for clinicians and researchers regarding which interventions are most likely to help older persons remain robust and independent. This paper uses meta-analysis to assess effectiveness of primary care interventions for physical frailty among community-dwelling adults aged 60+ and provides an up-to-date synthesis of literature in this area.

### Methods

PubMed, CINAHL, Cochrane Register of Controlled Trials, and PEDro databases were searched, and RCTs, controlled pilot studies, or trials with similar study designs addressing frailty in the primary care setting among persons aged 60+ were chosen. Study data was abstracted following PRISMA guidelines, then meta-analysis was performed using the random effects model.

### Results

31 studies with a total of 4794 participants were analysed. Interventions using predominantly resistance-based exercise and nutrition supplementation seemed to improve frailty status versus control (RR = 0.62 (CI 0.48–0.79), $I^2$ = 0%). Exercise plus nutrition education also reduced frailty (RR = 0.69 (CI 0.58–0.82), $I^2$ = 0%). Exercise alone seemed effective in reducing frailty (RR = 0.63 (CI 0.47–0.84), $I^2$ = 0%) and improving physical performance (RR = 0.43 (CI 0.18–0.67), $I^2$ = 0%). Exercise alone also appeared superior to control in improving gait speed (SMD = 0.36 (CI 0.10–0.61, $I^2$ = 74%), leg strength (SMD = 0.61 (CI 0.09–1.13), $I^2$ = 87%), and grip strength (Mean Difference = 1.08 (CI 0.02–2.15), $I^2$ = 71%)

**Funding:** This study was funded by Ireland's Health Research Board (https://www.hrb.ie/) and St. Vincent's University Hospital (https://www.stvincents.ie/) under the Applied Partnership Awards, grant number APA-2016-1857 held by Principal Investigator Dr. Marie Therese Cooney (MTC). Prof. Roman Romero-Ortuno (RRO) is funded by Science Foundation Ireland (https://www.sfi.ie) under the 2018 President of Ireland Future Research Leaders Programme, grant number 18/FRL/6188. The funders had no role in study design, data collection and analysis, decision to publish, or preparation of the manuscript.

**Competing interests:** The authors have declared that no competing interests exist.

though a high degree of heterogeneity was observed. Comprehensive geriatric assessment (RR = 0.77 (CI 0.64–0.93), $I^2$ = 0%) also seemed superior to control in reducing frailty.

## Conclusion

Exercise alone or with nutrition supplementation or education, and comprehensive geriatric assessment, may reduce physical frailty. Individual-level factors and health systems resource availability will likely determine configuration of future interventions.

## Introduction

Frailty can be defined as a physiological state of vulnerability due to dysregulation in multiple physiological systems where an individual's functional capacity and resilience against external stressors is reduced, leading to greater rates of illness, disability, and death.[1–3] Frailty can affect persons of any age, frequently arising in specific contexts such as malnutrition or long-term disease states such as diabetes, chronic obstructive pulmonary disease,[4] chronic heart failure,[5] or HIV infection.[6–8] Whilst it is tempting to view frailty simply as an inevitable consequence of ageing that is characterised by an increased risk of poor health outcomes,[9] a distinction must be made between an individual's chronological and biological age, as some people may remain robust and disability-free even at advanced ages.[10–12] Individual-level factors such as resilience (physical and mental), external supports, and other forms of intrinsic capacity,[13] can help moderate frailty, and these also require consideration when examining pathways to alleviate frailty in clinical practice.[14,15]

Timely identification and interventions to address frailty will support older individuals to build resilience and live independently, but also help health systems use resources more efficiently in the context of growing life expectancy worldwide.[12,16–19] However, clinical circumstances, individual person factors, and resource constraints all affect identification and management of frailty, with the likelihood being that no "one size fits all" approach exists. [20,21] This is thematically supported by the systematic review and meta-analysis by Ekelund and colleagues, which found reductions in premature mortality among older adults with higher levels of physical activity, regardless of activity type or intensity.[22] Accordingly, evidence-based guidance is needed to aid clinicians in determining the most appropriate interventions. This may be hampered by the wide range of frailty measurement, operationalisation, and intervention types.[23–25]

Numerous frailty measures have been deployed across acute and primary care settings. [19,25–28] This diversity has arisen out of necessity as frailty incorporates numerous biological, functional, and psychosocial dimensions.[29] A widely used frailty operationalisation is the "physical frailty phenotype", proposed by Fried and colleagues in 2001 as a syndrome defined by exhaustion, unexplained weight loss, reduced handgrip strength, slow gait speed, and reduced physical activity.[3] As such, physical frailty is conceptualised as being closely related with sarcopenia, which denotes poor skeletal muscle mass and/ or function.[30] Therefore, measures of sarcopenia are often components of, or are closely related to, physical frailty. [31] There have been numerous adaptations of the original frailty phenotype reported in the literature.[32] Although physical frailty and disability overlap, the former is conceptualised as a pre-disability state[33] that may offer more possibilities for delaying or reversing the disabling process through interventions.

In this study, we examined the effectiveness of primary care interventions against physical frailty among community-dwelling older adults aged over 60 years. Our previous qualitative synthesis of the literature suggested that a combination of muscle strength training and protein supplementation was the most effective intervention to delay or reverse frailty, and the easiest to implement in primary care. However, the highly heterogeneous nature of the retrieved studies meant that a single meta-analysis could not be readily performed. To address that gap, we performed a new series of smaller meta-analyses, categorised by intervention and outcome measurement type, following the broad question: "What primary care interventions for community-dwelling older (60+) adults are superior to control in reducing frailty and its associated measures?"

## Methods

### Database searches, eligibility criteria and article selection

The terms ("primary care" or "community") and ("screening" or "intervention" or "integrated-care") and ("frailty" or "pre-frail") were used to search the PubMed, CINAHL, Cochrane Register of Controlled Trials, and PEDro databases for articles in English without restriction on publication date, and excluding reviews. The protocol for this meta-analysis was not prospectively registered. Two authors independently reviewed titles and abstracts, and selected articles to include. Disagreements were resolved at meetings with at least three authors present. Articles selected for meta-analysis spanned a publication timeframe from May 1996 to June 2019 (time of writing), with 4 articles published in 2009 or earlier, 13 published between the beginning of 2010 and the end of 2015, and 14 published between the beginning of 2016 and the end of June 2019.

Published randomised controlled clinical trials, controlled pilot studies, or trials with similar study designs were considered. Reviews, case studies, and abstract-only publications were excluded. Eligible articles reported results of trials of interventions that focused on treating, delaying, or reversing physical frailty in the primary care setting among persons aged 60+. Where data was not available from the published studies, we emailed authors directly to request primary data for inclusion. If we were unable to contact the authors after two attempts, the article was excluded. Risk of bias was assessed at study level by two authors.

### Transformation of data and meta-analysis

Data was entered into a data extraction spreadsheet; specific outcome measures were pre-specified by the research team and included Fried criteria or adaptations, and/ or sarcopenia-related physical performance indicators. In studies with multiple timepoints we selected the immediate post-intervention timepoint where data was reported. For cross-over trial designs, we selected the last timepoint before cross-over. Results presented as percentages (e.g. percentage of participants who improved) were transformed into absolute figures where needed.

For dichotomised outcomes we have presented results as risk ratios with 95% confidence intervals, and performed number needed to treat (NNT) analysis by calculating absolute risk reduction (ARR) using the formula NNT = 1/ARR, where ARR results from subtracting the experimental event ratio (EER) from the control event ratio (CER) (ARR = CER-EER). For studies reporting multiple sub-groups for certain measures, these were combined into one group for pair-wise comparison with control, as recommended by the Cochrane Collaboration guidelines.[34] Continuous outcome measurement results are presented as standardised mean differences (SMD) plus 95% confidence intervals. In some studies, change scores were reported as opposed to final measurements, e.g. mean changes in time taken to perform a test versus mean final measurements of the time taken to perform the same test. If comparison

with final measurements was required, these were included in the relevant meta-analyses and results presented as *unstandardized* mean differences and 95% confidence intervals, in the manner recommended by the Cochrane Collaboration.[34] Studies using change scores were included as sub-groups within their respective analyses for clarity. For articles reporting gait speed as mean number of seconds taken to walk the test distance, rather than speed, these values were entered as reported and transformed to negative values to correct for directionality (i.e. improvements reflected by a decrease in mean time taken). [35]

Meta-analysis was performed using *Review Manager* (*RevMan*) version 5.3 using the inverse variance method, and random effects model to account for expected differences such as intervention dose and population variance. Inconsistency among the studies in each analysis was quantified using the $I^2$ statistic. Risk of bias was assessed in the manner recommended by the Cochrane Collaboration.[34]

## Results

The search was initially completed in May 2018 and repeated at the end of June 2019. Thirty-one articles were included in this meta-analysis after the review process was completed (Fig 1). There was a total of 4794 participants at randomisation across the included studies, with a median study size of 100 participants, maximum size of 459, and minimum of 23. Characteristics of included studies are presented in S2 Table. We prepared a matrix of outcome measures used in each study to help group them together for meta-analysis in cases where two or more studies used the same, or conceptually similar, measure (S3 Table).

Two studies were included in the meta-analysis of frailty alongside those reporting Fried Criteria: Behm et al. used expanded criteria incorporating Fried's[36] and Luger et al. used SHARE-FI[35] which is based on a latent class approach to modified Fried's criteria. Studies by Kim et al.,[37] Ng et al.,[38] and Kwon et al.[39] examined the effect of exercise, nutrition supplementation or education, and combination interventions. Such data was treated as separate comparisons to control but analysed in discrete meta-analyses to avoid double-counting the shared control group.

Risk of bias summaries are presented for descriptive and exploratory purposes, and did not influence inclusion or weighting of studies in the meta-analysis (Fig 2).

### Effect of interventions on physical frailty status

12 studies assessed frailty status using Fried criteria or similar measures, reporting change in frailty status, or frailty prevalence. We grouped these by intervention, conducting meta-analysis wherever two or more studies used similar interventions. Interventions using exercise and nutrition supplementation favoured intervention over control (n = 2, RR = 0.62 (CI 0.48–0.79), $I^2$ = 0%), whilst nutrition alone did not seem effective (n = 2, RR = 0.91 (CI 0.63–1.33), $I^2$ = 72%). Exercise and nutrition education (n = 4, RR = 0.69 (CI 0.58–0.82), $I^2$ = 0%), exercise-only (n = 4, RR = 0.63 (CI 0.47–0.84), $I^2$ = 0%), and comprehensive geriatric assessment (CGA) (n = 3, RR = 0.77 (CI 0.64–0.93), $I^2$ = 0%) all seemed effective versus control (Fig 3). Number needed to treat analysis of data from these studies indicated an NNT of 4 for exercise and nutrition supplementation, 8 for exercise and nutrition education, 11 for exercise only, and 15 for CGA.

### Effect of interventions on frailty-associated physical performance measures

Several additional physical frailty-associated measures were reported: physical performance tests such as the short physical performance battery (SPPB) or the physical performance test (PPT), gait speed, leg strength, timed up and go (TUG), grip strength, single-leg balance,

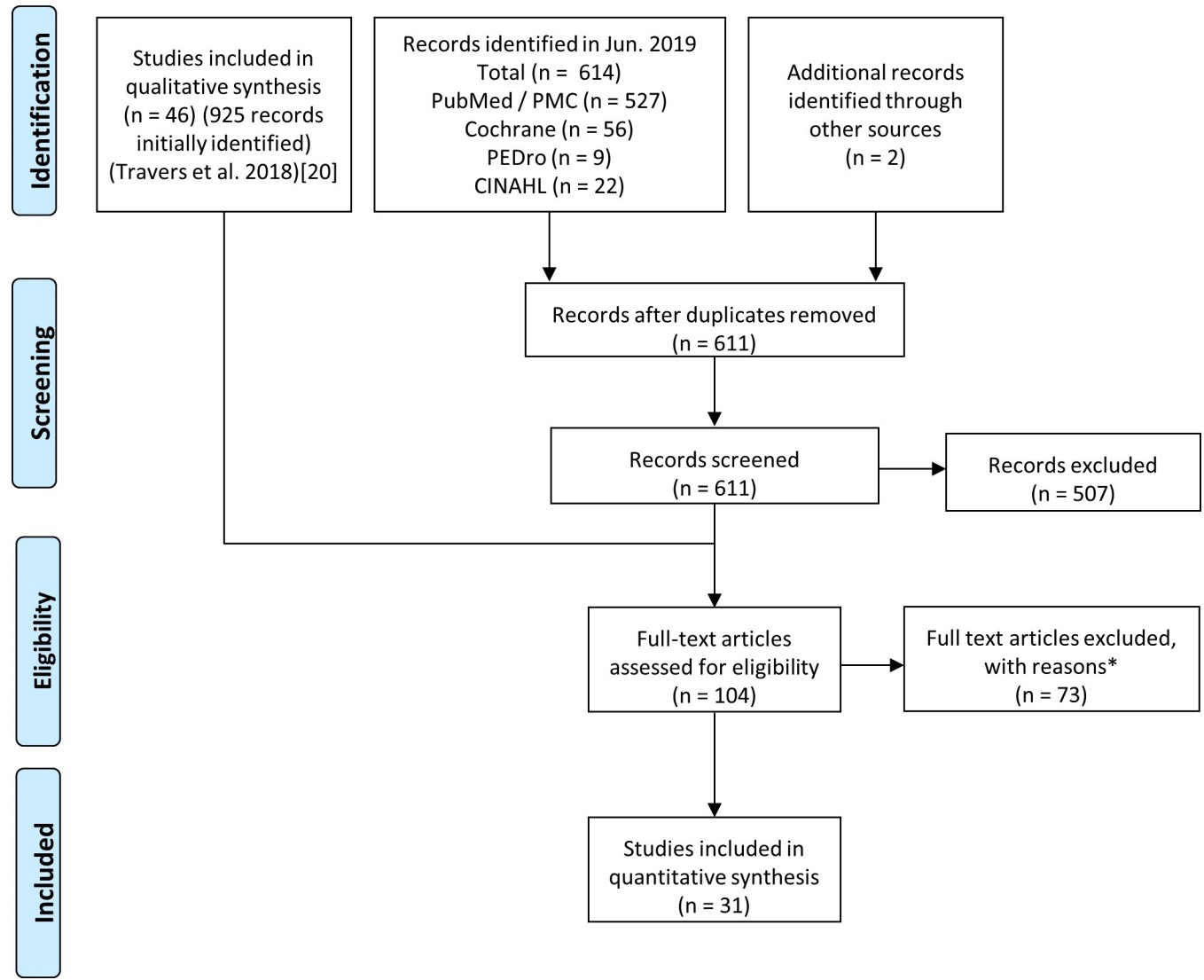

**Fig 1. Prisma flow diagram showing identification, screening, eligibility assessment, and inclusion of studies.** Reasons for exclusion of articles were: Ineligible outcome (13 studies); ineligible comparator (21), other (not yet published, irrelevant, not an intervention etc.) (34), unable to contact authors (4), authors contacted but primary data no longer accessible (1).

functional reach, and chair stands.[40] Four studies used physical performance as an outcome measure and employed exercise interventions, which seemed more effective than control (n = 4, SMD = 0.43 (CI 0.18–0.67), $I^2$ = 3%). Fifteen studies included gait speed as an outcome measure. Interventions using exercise and nutrition supplementation (n = 2, SMD = 0.26 (CI -0.19–0.71), $I^2$ = 52) were not superior to control, whilst exercise alone (n = 12, SMD = 0.36 (CI 0.10–0.61), $I^2$ = 74%) was superior although the results had a substantial degree of statistical heterogeneity. Nutrition only (n = 3, SMD = -0.01 (CI -0.26–0.24, $I^2$ = 0%) and exercise and nutrition education (n = 2, Mean Difference = 0.04 (CI -0.09–0.18), $I^2$ = 0%) were not superior to control (Fig 4).

Eleven studies included leg strength as a frailty-associated measure. Here, exercise and nutrition supplementation (n = 2, SMD = 0.21 (CI -0.26–0.67), $I^2$ = 54%) did not favour intervention over control, nor did nutrition supplementation alone (n = 3, SMD = -0.05 (CI -0.31–

**Risk of bias**

| Study | A | B | C | D | E | F | G |
|---|---|---|---|---|---|---|---|
| Akihiro et al. 2018 | − | ? | ? | ? | + | ? | + |
| Behm et al. 2016 | + | + | ? | + | + | ? | + |
| Binder et al. 2002 | + | + | ? | + | ? | ? | + |
| Brown et al. 2000 | ? | ? | ? | ? | ? | ? | + |
| Cesari et al. 2015 | + | + | ? | + | ? | ? | + |
| Chan et al. 2012 | + | + | ? | + | + | ? | + |
| Chan et al. 2017 | + | + | ? | + | + | − | + |
| Chen et al. 2019 | + | + | − | + | + | − | + |
| Clegg et al. 2014 | + | + | ? | ? | + | + | + |
| Coelho-Junior et al. 2019 | + | + | − | − | + | ? | + |
| Daniel et al. 2012 | ? | ? | ? | ? | ? | ? | ? |
| Fairhall et al. 2012 | + | + | ? | ? | + | − | ? |
| Hildreth et al. 2013 | + | ? | + | + | ? | ? | + |
| Jacobsen et al. 2012 | + | + | + | + | + | + | + |
| Kim and Lee 2013 | + | + | ? | + | + | + | + |
| Kim et al. 2015 | + | + | ? | + | ? | − | + |
| Kwon et al. 2015 | + | ? | ? | + | ? | ? | + |
| Li et al. 2010 | ? | ? | ? | + | + | ? | + |
| Liu et al. 2017 | − | ? | ? | + | ? | ? | ? |
| Luger et al. 2016 | + | ? | ? | + | + | ? | + |
| Mazya et al. 2018 | + | ? | ? | + | + | + | + |
| Muller et al. 2006 | + | ? | + | ? | + | ? | + |
| Nagai et al. 2018 | + | ? | ? | ? | + | + | ? |
| Ng et al. 2015 | + | + | ? | + | + | + | + |
| Oh et al. 2017 | ? | ? | ? | ? | ? | ? | ? |
| Seino et al. 2017 | + | ? | ? | + | + | ? | + |
| Serra-Prat et al. 2017 | + | + | ? | ? | + | + | + |
| Tarazona et al. 2016 | ? | ? | ? | + | + | + | + |
| Wolf et al. 1996 | + | ? | ? | + | ? | ? | + |
| Yamada et al. 2012 | − | ? | ? | + | − | ? | + |
| Yoon et al. 2018 | ? | ? | ? | ? | ? | ? | + |

Risk of bias legend

(**A**) Random sequence generation (selection bias)

(**B**) Allocation concealment (selection bias)

(**C**) Blinding of participants and personnel (performance bias)

(**D**) Blinding of outcome assessment (detection bias)

(**E**) Incomplete outcome data (attrition bias)

(**F**) Selective reporting (reporting bias)

(**G**) Other bias

**Fig 2. Risk of bias assessment of included studies.** Showing risk of bias as either low (green), unclear (yellow), or high (red) for included studies, for random sequence generation, allocation concealment, blinding of participants and personnel, blinding of outcome assessment, incomplete outcome data, selective reporting, or other bias.

0.21), $I^2 = 0\%$), or hormone supplementation (n = 2, SMD = -0.29 (CI -1.01–0.43), $I^2 = 77\%$), and whilst exercise only (n = 8, SMD = 0.61 (CI 0.09–1.13), $I^2 = 87\%$) favoured intervention over control a substantial degree of heterogeneity was observed (Fig 5).

Seventeen studies measured grip strength as a frailty-associated outcome, using a variety of interventions. Only exercise alone favoured intervention over control in this case, and there was a substantial degree of heterogeneity (n = 9, Mean Difference = 1.08 (CI 0.02–2.15, $I^2 = 71\%$). The other interventions here did not appear effective versus control. These were: Exercise plus nutrition education (n = 4, Mean Difference = 0.71 (CI -0.17–1.59, $I^2 = 0$), nutrition supplementation (n = 3, SMD = -0.10 (CI -0.43–0.23), $I^2 = 33\%$), and hormone supplementation (n = 3, Mean Difference = 0.10 (CI -1.13–1.34), $I^2 = 0\%$) (Fig 6).

Thirteen studies included the timed up and go test as an outcome measure, but in our meta-analysis none of the interventions tested were superior to control. These were: nutrition supplementation (n = 3, SMD = -0.04 (CI -0.31–0.23), $I^2 = 0\%$), exercise alone (n = 8, SMD = -0.47 (CI -0.98–0.05), $I^2 = 82\%$), and exercise plus nutrition education (n = 3, Mean Difference = -0.42 (CI -1.06–0.23), $I^2 = 53\%$) (S1 Fig). Seven studies used single-leg balance as an outcome measure and employed exercise and nutrition education (n = 4, Mean Difference = 3.88 (CI -1.72–9.48), $I^2 = 71\%$) and exercise only (n = 4, Mean Difference = 0.04 (CI -2.98–3.06), $I^2 = 60\%$), which were not superior to control (S2 Fig). Three studies tested the effect of exercise on functional reach, which was not found to be superior to control (n = 3, SMD = 0.19 (CI -0.10–0.48), $I^2 = 0\%$) (S2 Fig). Lastly, exercise was not superior to control for three studies using chair stands as an outcome measure, and these studies displayed a high degree of heterogeneity (n = 3, SMD = 0.63 (CI -1.02–2.29), $I^2 = 92\%$) (S2 Fig).

## Discussion

Our results suggest that physical frailty and related measures may be improved, mainly through resistance-based exercise. Exercise can confer functional and cognitive benefits for older individuals, even in the presence of comorbidities,[41,42] and our findings provide an up-to-date synthesis of the literature in support of this concept.

In some cases, exercise alone appeared to improve frailty-associated measures such as gait speed or leg strength (albeit with a high degree of heterogeneity) whilst exercise plus nutrition supplementation did not. An additive effect might have been expected, but we only retrieved two studies utilising this combination of strategies therefore the reason for this effect remains unclear. Further exploration of the underlying causes of this variation (e.g. participant heterogeneity, nutritional status at time of recruitment, dosage, compliance etc.) should take precedence over a quick interpretation that nutrition supplementation plus exercise is not a useful intervention. The range of possible dietary supplementation measures is also likely to be a confounding factor.[43] Although we retrieved only a few trials that examined the effect of nutrition alone in addressing frailty, and did not find it superior to control in improving any measures, it remains a potential avenue for further research. Indeed, poor nutritional status is associated with onset of frailty,[44] and intervention against the background levels of malnutrition commonly seen among older and frail individuals may at least act as an adjunct to support those individuals to make the best possible progress as they move through care and rehabilitation.[45,46]

**A**

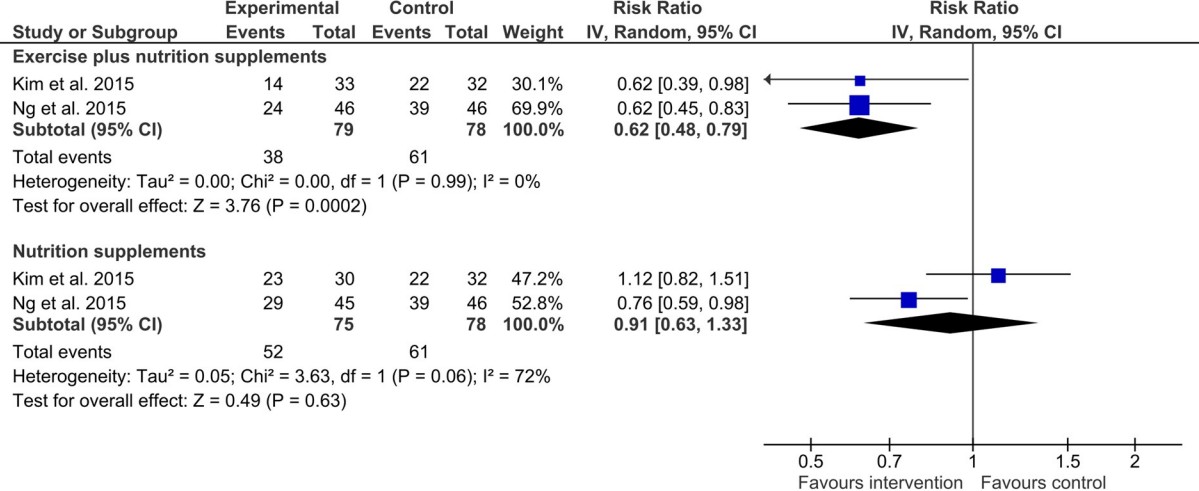

**B**

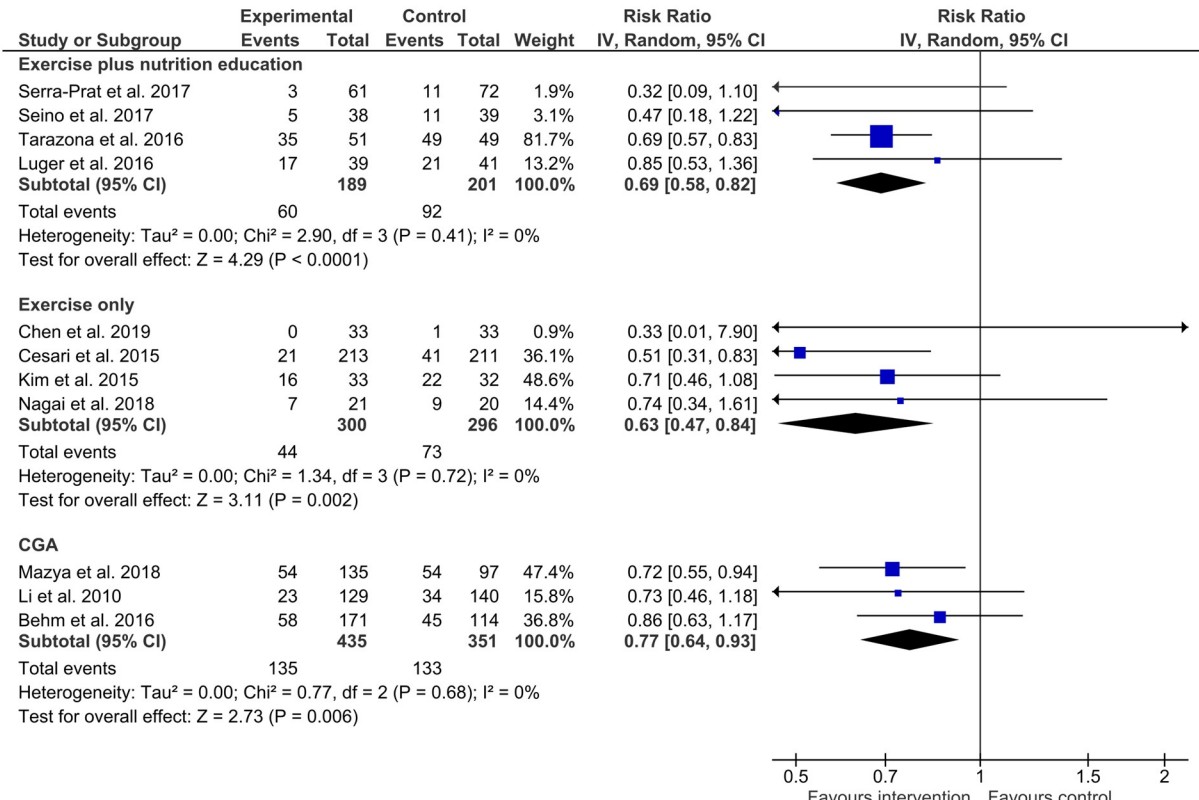

**Fig 3. Effect of interventions on frailty status.** A) Shows Risk ratio (RR, square data markers) with 95% confidence intervals (CI, horizontal lines) for meta-analysis of change in frailty status (dichotomised to not frail vs. frail; events indicate individuals whose status did not change) for interventions using exercise plus nutrition supplements or nutrition supplements alone. B) Shows RR with 95% CI for meta-analysis of change in prevalence of frailty (dichotomised to not frail vs. frail; events indicate frail individuals) for interventions using exercise plus nutrition education, exercise only, and comprehensive geriatric assessment. Size of data markers indicates study weighting in random effects meta-analysis. Diamond-shaped data markers indicate overall RR and CI for each grouping of interventions. I² statistic is reported for each group. Left side favours intervention.

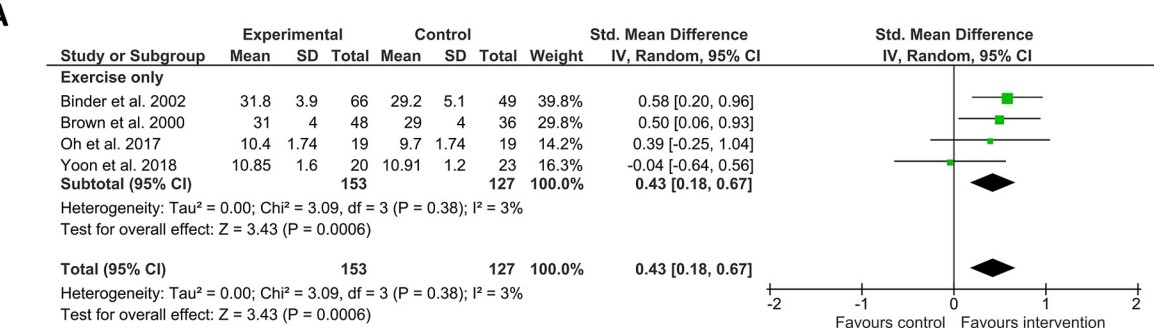

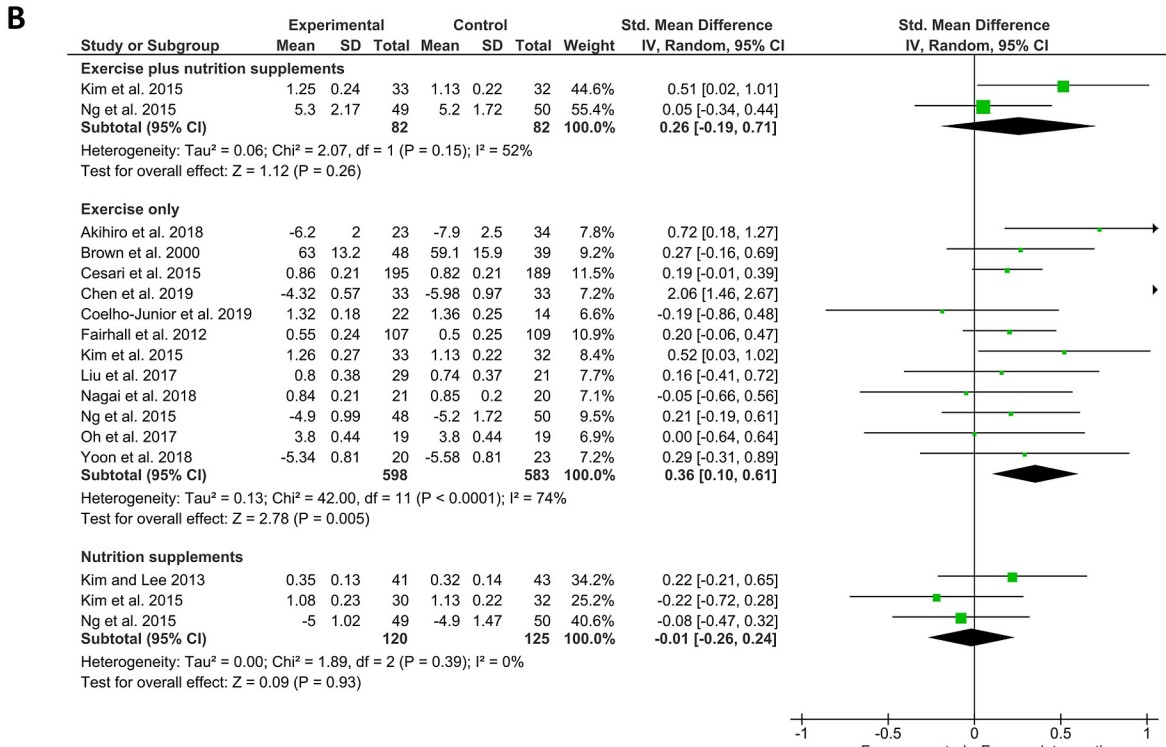

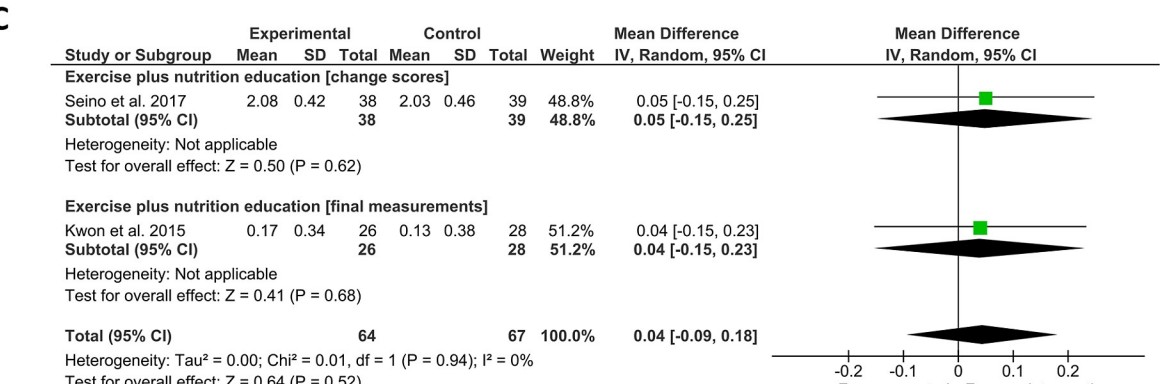

**Fig 4. Effect of interventions on physical performance and gait speed.** A) Shows SMD (square data markers) with 95% CI (horizontal lines) for meta-analysis of mean scores ± SD for physical performance tests for interventions using exercise. B) Shows SMD with 95% CI for meta-analysis of gait speed as mean time taken to walk a set distance (measured in seconds) or mean speed ± SD (measured in metres

per second) for interventions using exercise plus nutrition supplements, exercise only, or nutrition supplements only. C) Shows Mean Difference with 95% CI for meta-analysis of gait speed (measured metres per second), reported either as change scores or as final measurements ± SD, separated into sub-groups for clarity, for interventions using exercise plus nutrition education. Size of data markers indicates study weighting in random effects meta-analysis. Diamond-shaped data markers indicate overall SMD or Mean Difference with CI for each grouping of interventions. I² statistic is reported for each group. Right side favours intervention.

Since frailty by nature is a complex construct comprising multiple interacting dimensions of capacity, resilience, and incapacity,[3,29] it is unsurprising that some interventions purporting to address "frailty" as a phenomenon showed little effect on single-dimension frailty-associated measures. For example, endurance training focusing on walking and lower body strength could not reasonably be expected improve handgrip strength. It could, however, be expected to improve criteria such as exhaustion, gait speed, or physical activity. Changes in such measures cannot themselves be interpreted as constituting improvements in overall

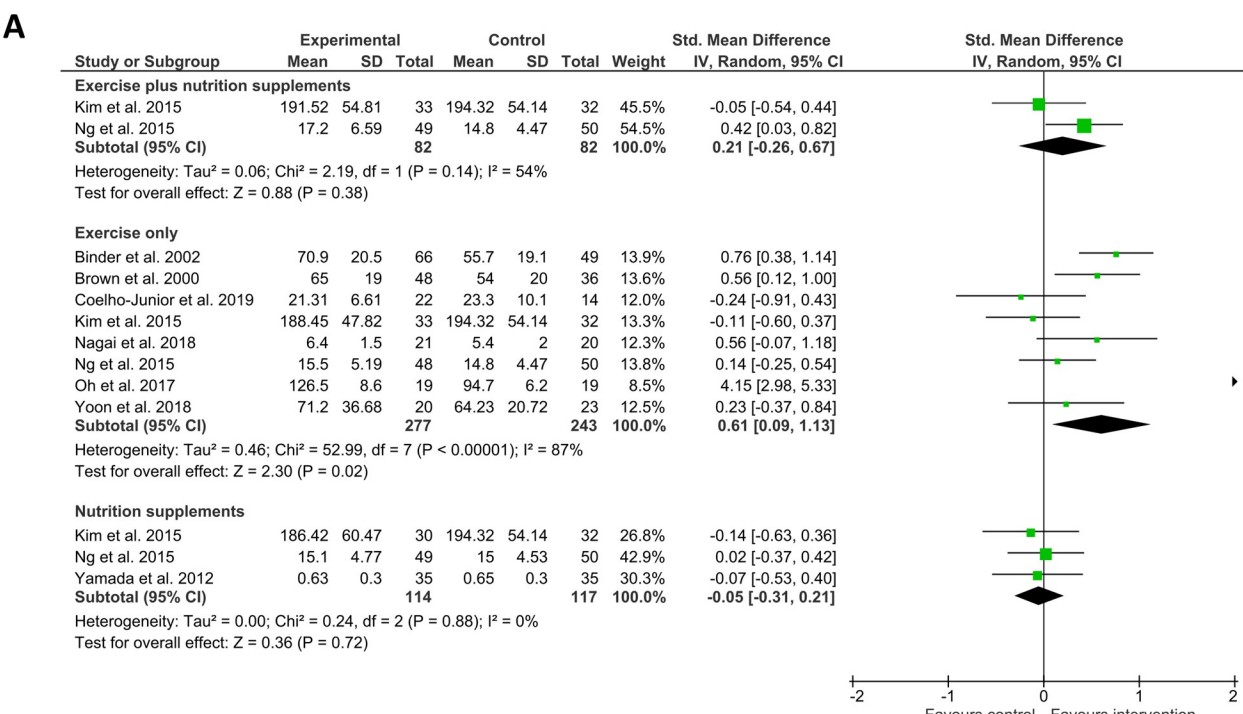

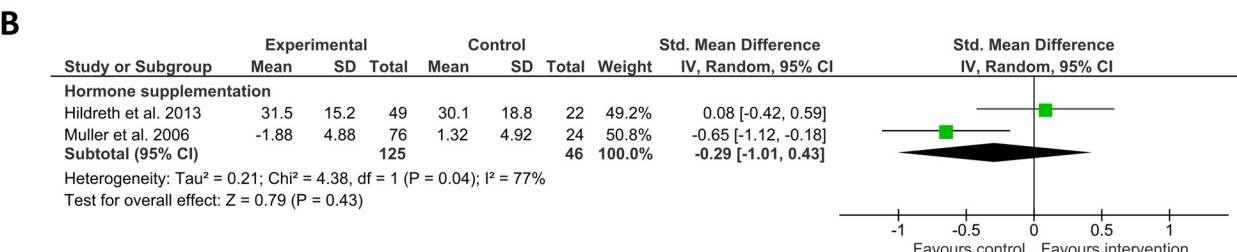

**Fig 5. Effect of interventions on leg strength.** A) Shows SMD (square data markers) with 95% CI (horizontal lines) for meta-analysis of mean scores ± SD for leg strength (measured in kilogrammes, kilogramme metres, feet per pound, Newtons, or Newton metres) for interventions using exercise plus nutrition supplements, exercise only, and nutrition supplements only. B) Shows SMD with 95% CI for meta-analysis of leg strength reported as mean change from baseline ± SD (measured in Kilogrammes or Newton Metres), for interventions using hormone supplementation. Size of data markers indicates study weighting in random effects meta-analysis. Diamond-shaped data markers indicate overall SMD with CI for each grouping of interventions. I² statistic is reported for each group. Right side favours intervention.

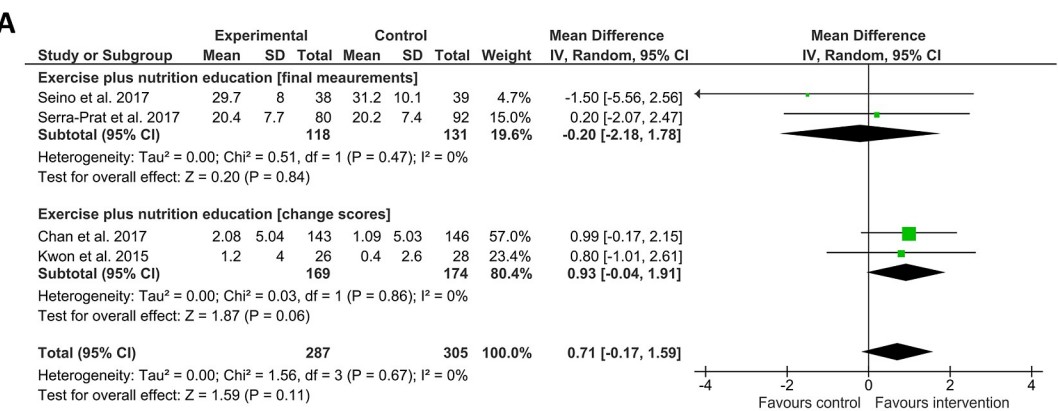

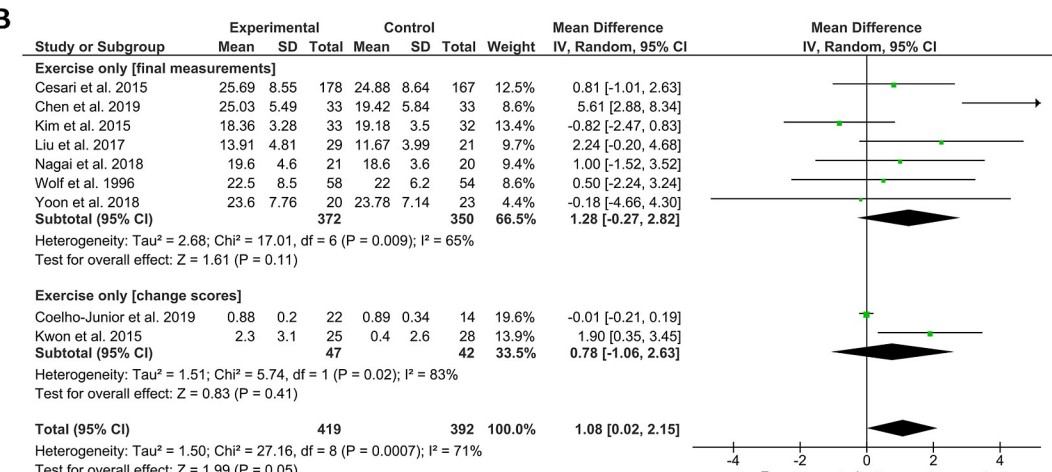

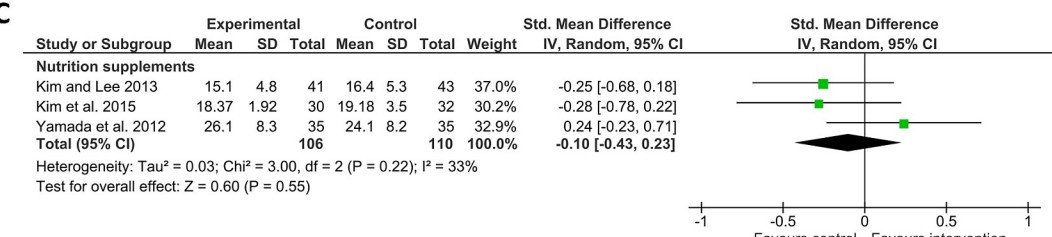

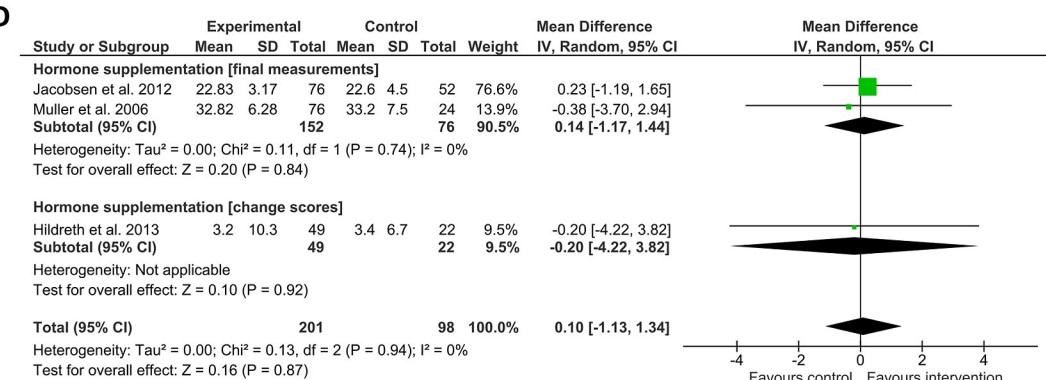

**Fig 6. Effect of interventions on grip strength.** A, B, D) Show Mean Difference (square data markers) with 95% CI (horizontal lines) for meta-analysis of grip strength (measured in kilogrammes), reported either as change scores or as final

measurements ± SD, separated into sub-groups for clarity, for interventions using exercise plus nutrition education, exercise only, or hormone supplementation. C) Shows SMD with 95% CI for meta-analysis of mean grip strength (measured in kilogrammes) ± SD, for interventions using nutrition supplements. Size of data markers indicates study weighting in random effects meta-analysis. Diamond-shaped data markers indicate overall SMD or Mean difference with CI for each grouping of interventions. $I^2$ statistic is reported for each group. Right side favours intervention.

frailty, as they only represent one component in isolation. Caution must therefore be exercised in interpreting results in these cases, and attention must still be paid to the individual's underlying risk factors associated with frailty. That notwithstanding, our results showing effects of interventions such as exercise in improving leg strength and gait speed are somewhat consistent with previous evidence syntheses from other groups that examined interventions against sarcopenia[47]–itself linked to many of the frailty-associated measures included in our analysis.[30] Some interventions may also simply not yet have a well-developed literature base: although hormone supplementation has been shown to improve strength and muscle power in settings outside of frailty,[48] there was only sufficient data among frailty-associated measures for us to meta-analyse its effect on leg strength and grip strength. No overall effect was shown, although those studies did report gains in other measures such as upper body strength and body composition.[49,50] Synergistic benefits might still be gained from hormone supplementation in tandem with targeted functional exercise interventions against frailty. It may be the case that future combinatorial interventions will yield the greatest benefits.

## Heterogeneity among studies

$I^2$ statistics for our meta-analyses were generally low to moderate,[34] but when comparing nutrition interventions' effect on changing frailty status, exercise-only interventions' effect on leg strength and gait speed, effect of exercise and nutrition education, or exercise alone, on single-leg balance, and effect of exercise on chair stands, the $I^2$ was considerable, ranging from 71% to 92%. We had conducted the meta-analyses using the random effects model with the aim of producing more conservative levels of significance for the computed effect estimates [34] however we explored whether small-study effects might have been influencing the results by re-analysing using the fixed-effects model for the three analyses where a significant intervention effect was observed in the presence of heterogeneity greater than 60%. For the analysis of the effects of exercise alone on gait speed or leg strength the intervention effect estimate remained significant under the fixed-effects model, but for the analysis of the effect of exercise alone on grip strength, the effect estimate became non-significant (S3 Fig). In the latter instance, the apparent positive intervention effect may have been due to the small-study effect. This does not preclude the presence of small-study effects in the other two analyses, however it was not likely the only influencing factor.

In general, observed heterogeneity may have been due to diversity between approaches, since we did not narrow our searches within interventions (e.g. resistance training versus bodyweight, aerobic exercise etc.) but rather sought to examine if any intervention at all was a useful strategy. Dosage or adherence could also have influenced heterogeneity in outcomes–for example where self-directed exercise was encouraged in conjunction with the prescribed interventions, as opposed to only using a fixed regime. The frailty status of participants at the beginning of the studies is also likely to have been a source of heterogeneity to some degree: For example, the Fried criteria specify characteristics such as slowness, weight loss etc., which if present categorise a person as robust, pre-frail, or frail, and in many cases participants were classified this way at baseline, with pre-frailty often a selection criteria for trials. It is, however, difficult to identify each individual's capacity for improvement or decline, and a seemingly

homogenous group of participants categorised as "pre-frail" might nevertheless each experience markedly different outcomes. A mixture of clinical and methodological heterogeneity is the most likely case, and incorporation of future studies in the area will aid in exploring the degree to which each of these exerts an influence.

The small number of studies in each outcome measure grouping also precluded our exploring heterogeneity using funnel plots in most cases.[51] However, we were able to do so for our meta-analysis of the effect of exercise on gait speed (S4 Fig), finding some degree of asymmetry around the overall effect line. For exploratory purposes, we excluded individual studies from the analysis and found that one study, Chen et al.,[52] was responsible for introducing the observed asymmetry (although we note additionally that the overall intervention effect remained significant even when that study was removed from the analysis). Participants in Chen and colleagues' study were not blinded as to their allocated groups, which may have introduced some degree of performance bias (acknowledged as a key a difficulty in designing exercise interventions[53]).

## Suggestions for further research

We recommend that further studies should aim to delineate the optimal exercise interventions to delay or improve frailty status. Giné-Garriga and colleagues' 2014 systematic review and meta-analysis noted that there will certainly be differences in the type of interventions suitable for non-frail older adults compared with those who are frail–for example the latter may require shorter-duration, performance-based sessions.[54] Future research should also focus on contextual factors supporting frailty interventions, such as health resources provisioning,[55] linkages between acute and long-term care services,[56] configuration of care packages,[57] and public and patient involvement in co-designing interventions.[58] There is also some incompatibility between frailty measures themselves, necessitating caution in comparing outcomes. [24] Coordinated international movements towards a unifying set of frailty measures may be challenging, but will greatly aid future meta-analyses to clarify best-practice interventions.[2] Our study may be interpreted as summarising the evidence related to the physical frailty phenotype, but the results should not be extrapolated to other frailty definition approaches such as the frailty index,[59] which defines frailty according to an accumulation of deficits model which often includes more advanced disability items. Lastly, several recent publications have reported successful results of multifactorial, multidisciplinary approaches simultaneously targeting physical activity, nutrition, polypharmacy, depression, cognitive function, and other factors.[60–64] These studies were beyond the scope of our meta-analysis due to the highly diverse components in each intervention, but this holistic approach certainly warrants further attention by researchers.

## Implications for clinical practice

Resistance-based exercise could be employed alone, or in conjunction with nutrition supplementation or education, to help maintain functional capacity during the ageing process. [65,66] The majority of studies included in our meta-analysis featured resistance-based exercise, as performed using weights, elastic bands, or body weight. These were done alone or alongside aerobic or coordination and balance components. The ideal type, duration, and frequency of exercise will need to be tailored on a per-person basis, and such programming will likely feature a high degree of flexibility in configurations. As a starting point, our previous systematic review suggested generally higher effectiveness of resistance-based exercises compared to other forms of activity,[20] with the largest relative demonstrated effectiveness from a combinatorial programme of mixed exercises incorporating resistance.[67] Implementers may be

guided by the existing syntheses of the current literature, for example as summarised by Bray and colleagues[68] with training 2–3 times weekly using multi-component exercise programs featuring aerobic, resistance, balance, and flexibility, commencing at moderate intensity and gradually increasing to moderate-vigorous levels.[69–71] Frost and colleagues' recent systematic review and meta-analysis also highlighted a need to explore the role of exercise in groups in determining effectiveness of interventions.[72] There is also a small but growing evidence base indicating relatively greater effectiveness of multi-domain over mono-domain interventions.[73]

Dietary changes might include appropriately-considered uptake of plant-based proteins, or other protein sources such as milk, fish, eggs, chicken, or commercial formula nutrient drinks, whilst nutritional education could focus on improvements in dietary variety, particularly with regard to protein intake. Initiatives such as group activities, goal setting, and checklists may also help improve nutritional status by encouraging consumption of a properly balanced diet. [35,74]

Three studies included CGA or CGA-like strategies as an intervention,[36,75,76] incorporating tools such as preventive home visits, multi-professional group meetings to discuss patients' care, and treatments being prescribed based on individual need. Our meta-analysis suggests that this tailored approach is successful in reducing frailty prevalence but it is difficult to ascertain whether a specific component was responsible for improvement, or if there was a combinatorial effect, due to the varied needs of the individuals taking part in those studies.

## Limitations

Our database searches were limited to literature published in English. This may be a source of bias towards selection of only positive results, which are more likely to be reported in international English-language journals whilst negative results may more commonly appear in local-language journals.[34] In this case, limitations of time and language expertise precluded full coverage. It was difficult to determine risk of bias regarding selective reporting of results within studies since original trial protocols were not available for many of the retrieved articles. However, this risk could be somewhat attenuated by our focus on commonly-reported frailty indices and associated measures.

We could only include a small number of studies in our meta-analyses since, for any given intervention, they frequently did not employ the same outcome metrics. This variability is a common feature noted by other systematic reviews and meta-analyses.[54,72] We also chose to include some non-RCT study designs such as controlled pilot studies. This decision was made pragmatically to incorporate non-randomised trials which nevertheless were of sufficient quality and had usable data.[77] Our meta-analyses were also limited to cases where two or more studies used the same effect measures, reflecting the current fragmented status of the literature around frailty interventions as a whole.[20,25] The difficulty in comparisons between frailty indices is also not a minor challenge for analysts,[24] and it must be noted that studies may have shown significant effects in reducing frailty[78] but could not be compared with others since they used unique indicators developed specifically for those studies.[59]

A possible limitation is that exercise/ physical activity is included in the definition of physical frailty, and as such exercise intervention may improve the definition, but not necessarily the person. This has been a criticism in the literature (especially when using self-reported physical activity[79]) and so well designed clinical trials are still necessary to remove this possible source of bias that may be present in some included studies.

## Conclusions

Interventions using predominantly resistance-based exercise and nutrition supplementation, exercise and nutrition education, exercise alone, and CGA, appear effective in improving frailty status among community-dwelling adults aged 60 years or older. Such interventions may also improve frailty-associated measures of physical capacity. Future research should focus on exploring the optimal configurations for these interventions. Insufficient evidence is available at time of writing to support the use of interventions such as hormone supplementation in addressing frailty status or frailty-associated measures of physical function, and future analyses should seek to incorporate any forthcoming studies in this area in order to create meaningful syntheses of their effect. Lastly, there may be value in expanding research comparing the multiple diverse frailty indices that exist, and studying their individual components, to aid future meta-analyses. Although only a small number of articles was retrieved, our meta-analysis follows our previous systematic review[20] in presenting an up-to-date view of the available information at the time of writing, and may guide clinicians and researchers wishing to develop and refine interventions against frailty in the primary care setting.

## Supporting information

**S1 Table. PRISMA checklist.**
(PDF)

**S2 Table. Characteristics of included studies.** Showing study identifier, methods, participant information, interventions, outcomes, and citation for the studies included in the meta-analysis.
(XLSX)

**S3 Table. Matrix of relevant outcome measures in included studies.** Showing intervention type, study identifier, and outcome measures used in the studies included in the meta-analysis.
(XLSX)

**S1 Fig. Effect of interventions on timed up and go test.** A) Shows SMD (square data markers) with 95% CI (horizontal lines) for meta-analysis of mean ± SD time taken to perform timed up and go test (measured in seconds), for interventions using nutrition supplements or exercise alone. B) Shows Mean Difference with 95% CI for meta-analysis of time taken to perform TUG (measured in seconds), reported either as change scores or as final measurements ± SD, separated into sub-groups for clarity, for interventions using exercise plus nutrition education. Size of data markers indicates study weighting in random effects meta-analysis. Diamond-shaped data markers indicate overall SMD or Mean difference with CI for each grouping of interventions. $I^2$ statistic is reported for each group. Left side favours intervention.
(TIF)

**S2 Fig. Effect of interventions on single-leg balance, functional reach, and chair stand test.** A, B) Show Mean Difference (square data markers) with 95% CI (horizontal lines) for meta-analysis of single-leg balance (measured in seconds) reported either as change scores or as final measurements ± SD, separated into sub-groups for clarity, for interventions using exercise plus nutrition education and exercise only. C) Shows SMD with 95% CI for meta-analysis of mean scores (measured in centimetres or inches) for functional reach ± SD, for interventions using exercise. D) Shows SMD with 95% CI for meta-analysis of mean scores in chair stand test (measured as number of stands in 30 seconds or time taken to rise 5 times from sitting) ± SD for interventions using exercise. Size of data markers indicates study weighting in

random effects meta-analysis. Diamond-shaped data markers indicate overall SMD or Mean difference with CI for each grouping of interventions. $I^2$ statistic is reported for each group. Right side favours intervention.
(TIF)

**S3 Fig. Comparison of random effects model vs. fixed effects model for meta-analysis of exercise interventions' effect on gait speed, leg strength, and grip strength.** Left side shows meta-analysis using Random Effects model; right side shows meta-analysis using Fixed Effects model. A) Shows effect (SMD) of exercise interventions on gait speed, B) shows effect (SMD) of interventions on leg strength, and C) shows effect (Mean Difference) of interventions on grip strength. Size of data markers indicates study weighting in meta-analysis. Diamond-shaped data markers indicate overall SMD or Mean difference with CI for each grouping of interventions. $I^2$ statistic is reported for each group. Right side favours intervention.
(TIF)

**S4 Fig. Funnel plot of results of exercise interventions' effect on gait speed.** Open diamond shapes represent individual studies in the meta-analysis. Y axis shows SE of the SMD; X axis shows SMD. Blue dashed vertical line represents overall effect. Left panel shows funnel plot including the study by Chen et al. (2019); right panel shows funnel plot excluding Chen et al.
(TIF)

## Acknowledgments

Dr. Marie Therese Cooney is the Principal Investigator of the "SAFE: Systematic Approach to improving care for Frail Elderly patients research programme", funded by the HRB. The authors declare that there are no conflicts of interest. We gratefully acknowledge the assistance of the authors who helpfully provided primary data from their work to include in our meta-analysis.

## Author Contributions

**Conceptualization:** Stephen H. -F. Macdonald, John Travers, Éidín Ní Shé, Jade Bailey, Roman Romero-Ortuno, Diarmuid O'Shea, Marie Therese Cooney.

**Data curation:** Stephen H. -F. Macdonald, Jade Bailey, Michael Keyes, Marie Therese Cooney.

**Formal analysis:** Stephen H. -F. Macdonald, John Travers, Jade Bailey, Marie Therese Cooney.

**Funding acquisition:** Marie Therese Cooney.

**Investigation:** Marie Therese Cooney.

**Methodology:** Stephen H. -F. Macdonald, Éidín Ní Shé, Roman Romero-Ortuno, Marie Therese Cooney.

**Project administration:** Marie Therese Cooney.

**Supervision:** Roman Romero-Ortuno, Marie Therese Cooney.

**Validation:** John Travers, Jade Bailey, Michael Keyes, Marie Therese Cooney.

**Writing – original draft:** Stephen H. -F. Macdonald, John Travers, Éidín Ní Shé, Jade Bailey, Roman Romero-Ortuno, Michael Keyes, Diarmuid O'Shea, Marie Therese Cooney.

**Writing – review & editing:** Stephen H. -F. Macdonald, John Travers, Éidín Ní Shé, Jade Bailey, Roman Romero-Ortuno, Diarmuid O'Shea, Marie Therese Cooney.

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
