## [Decision Letter · Decision Letter 0]

13 Nov 2019

PONE-D-19-25888

Primary care interventions to address physical frailty among community-dwelling adults aged 60 years or older: A meta-analysis

PLOS ONE

Dear Dr. Macdonald,

Thank you for submitting your manuscript to PLOS ONE. After careful consideration, we feel that it has merit but does not fully meet PLOS ONE’s publication criteria as it currently stands. Therefore, we invite you to submit a revised version of the manuscript that addresses the points raised during the review process.

We would appreciate receiving your revised manuscript by Dec 28 2019 11:59PM. To enhance the reproducibility of your results, we recommend that if applicable you deposit your laboratory protocols in protocols.io, where a protocol can be assigned its own identifier (DOI) such that it can be cited independently in the future. For instructions see: http://journals.plos.org/plosone/s/submission-guidelines#loc-laboratory-protocols

We look forward to receiving your revised manuscript.

Kind regards,

Jose Vina

Academic Editor

PLOS ONE

Journal Requirements:

2. Please ensure that you have addressed all items recommended in the PRISMA checklist including identifying the study as a meta-analysis or systematic review in the title.

Reviewers' comments:

Reviewer's Responses to Questions

**Comments to the Author**

1. Is the manuscript technically sound, and do the data support the conclusions?

Reviewer #1: Yes

2. Has the statistical analysis been performed appropriately and rigorously? 

Reviewer #1: Yes

3. Have the authors made all data underlying the findings in their manuscript fully available?

Reviewer #1: Yes

4. Is the manuscript presented in an intelligible fashion and written in standard English?

Reviewer #1: Yes

5. Review Comments to the Author

Reviewer #1: This manuscript provides a complete meta-analysis on the interventions to treat physical frailty in adults over 60 years of age, carrying out a thorough study on the available bibliography and carefully selecting the data to be analyzed.

First of all I want to congratulate the researchers for the work done. As a proposal for improvement, I suggest that the time frame of the publications selected for analysis be included in the methodology section, between lines 94 and 139.

6. PLOS authors have the option to publish the peer review history of their article (what does this mean?). If published, this will include your full peer review and any attached files.

Reviewer #1: No

---

## [Author Response · Author response to Decision Letter 0]

15 Nov 2019

We thank the editors and reviewers for their consideration of our manuscript, and hereby submit a revised manuscript in track changes and clean versions plus a "response to reviewers" letter. We have made the following changes as requested:

1) We have updated the formatting style to conform to PLOS guidelines, and renamed supporting material files accordingly.

2) We have updated the PRISMA checklist to address all items, and we confirm that the study is labelled as a meta-analysis in the title, “Primary care interventions to address physical frailty among community-dwelling adults aged 60 years or older: A meta-analysis.”

3) We have addressed the reviewer’s comment, “As a proposal for improvement, I suggest that the time frame of the publications selected for analysis be included in the methodology section, between lines 94 and 139,” by adding a sentence to the methodology section, which can be seen in the revised manuscript, commencing at line 100.

---

## [Decision Letter · Decision Letter 1]

24 Jan 2020

Primary care interventions to address physical frailty among community-dwelling adults aged 60 years or older: A meta-analysis

PONE-D-19-25888R1

Dear Dr. Macdonald,

We are pleased to inform you that your manuscript has been judged scientifically suitable for publication and will be formally accepted for publication once it complies with all outstanding technical requirements.

With kind regards,

Jose Vina

Academic Editor

PLOS ONE

Additional Editor Comments (optional):

Reviewers' comments:

Reviewer's Responses to Questions

**Comments to the Author**

1. If the authors have adequately addressed your comments raised in a previous round of review and you feel that this manuscript is now acceptable for publication, you may indicate that here to bypass the “Comments to the Author” section, enter your conflict of interest statement in the “Confidential to Editor” section, and submit your "Accept" recommendation.

Reviewer #1: All comments have been addressed

2. Is the manuscript technically sound, and do the data support the conclusions?

Reviewer #1: Yes

3. Has the statistical analysis been performed appropriately and rigorously? 

Reviewer #1: Yes

4. Have the authors made all data underlying the findings in their manuscript fully available?

Reviewer #1: Yes

5. Is the manuscript presented in an intelligible fashion and written in standard English?

Reviewer #1: Yes

6. Review Comments to the Author

Reviewer #1: All the suggestions have been answered satisfactorily.

Congratulations on the work done in a growing line of research.

7. PLOS authors have the option to publish the peer review history of their article (what does this mean?). If published, this will include your full peer review and any attached files.

Reviewer #1: No

---

## [Editor Report · Acceptance letter]

27 Jan 2020

PONE-D-19-25888R1 

Primary care interventions to address physical frailty among community-dwelling adults aged 60 years or older: A meta-analysis 

Dear Dr. Macdonald:

I am pleased to inform you that your manuscript has been deemed suitable for publication in PLOS ONE. Congratulations! Your manuscript is now with our production department. 

With kind regards,

on behalf of

Dr. Jose Vina 

Academic Editor

PLOS ONE